# Analysis and Characterization of the Extracellular Vesicles Released in Non-Cancer Diseases Using Matrix-Assisted Laser Desorption Ionization/Mass Spectrometry

**DOI:** 10.3390/ijms25084490

**Published:** 2024-04-19

**Authors:** Antonella Maria Aresta, Nicoletta De Vietro, Carlo Zambonin

**Affiliations:** Department of Biosciences, Biotechnology and Environment, University of Bari “Aldo Moro”, Via E. Orabona 4, 70126 Bari, Italy; nicoletta.devietro@uniba.it (N.D.V.);

**Keywords:** MALDI-TOF/MS, extracellular vesicles, exosomes, diseases, diagnosis

## Abstract

The extracellular vesicles (EVs) released by cells play a crucial role in intercellular communications and interactions. The direct shedding of EVs from the plasma membrane represents a fundamental pathway for the transfer of properties and information between cells. These vesicles are classified based on their origin, biogenesis, size, content, surface markers, and functional features, encompassing a variety of bioactive molecules that reflect the physiological state and cell type of origin. Such molecules include lipids, nucleic acids, and proteins. Research efforts aimed at comprehending EVs, including the development of strategies for their isolation, purification, and characterization, have led to the discovery of new biomarkers. These biomarkers are proving invaluable for diagnosing diseases, monitoring disease progression, understanding treatment responses, especially in oncology, and addressing metabolic, neurological, infectious disorders, as well as advancing vaccine development. Matrix-Assisted Laser Desorption Ionization (MALDI)/Mass Spectrometry (MS) stands out as a leading tool for the analysis and characterization of EVs and their cargo. This technique offers inherent advantages such as a high throughput, minimal sample consumption, rapid and cost-effective analysis, and user-friendly operation. This review is mainly focused on the primary applications of MALDI–time-of-flight (TOF)/MS in the analysis and characterization of extracellular vesicles associated with non-cancerous diseases and pathogens that infect humans, animals, and plants.

## 1. Introduction

Intercellular communication is a fundamental process in the physiology of multicellular organisms, enabling the exchange of signals and resources essential to the body’s necessary functions and the maintenance of homeostasis [1]. Direct communication occurs through the gap junctions and synapses between physically adjacent cells [2], while systemic signals rely on alternative pathways, such as receptor–ligand interactions.

In both physiological and pathological conditions, another crucial mechanism of interaction among all living cells is the release of extracellular vesicles (EVs), a heterogeneous group of nanostructures formed by bilayer lipid membranes. The content of EVs comprises a variety of bioactive molecules that depend on the physiological state and the type of cell of origin, including lipids [3], nucleic acids [4], and proteins [5]. A comprehensive overview of the molecular composition (lipid, RNA, and proteins) of different subpopulations of EVs can be found in the manually curated database Vesiclepedia (http://microvesicles.org (accessed on 12 September 2023), while the exosome database ExoCarta (http://www.exocarta.org (accessed on 12 September 2016) lists 9769 proteins, 3408 mRNAs, 2838 miRNAs, and 1116 lipids identified in exosomes from multiple organisms. EVs circulate in biofluids such as blood, ascites, urine, saliva, and tears, carrying their cargo to local and/or distant body districts to exert roles in cellular communication, immune modulation, biomolecular transport, and physiological regulation.

Extracellular vesicles are generally classified into three primary categories, apoptotic bodies (ABs), microvesicles (MVs), and exosomes (EXOs) [6,7], based on their cell of origin (e.g., cancer, stem cell, NK cell, tissue, etc.), biogenesis, and size. ABs and MVs are larger EVs, with ABs formed during cell death via the apoptotic pathway and characterized by diameters ranging between 0.5 and 2 µm and MVs typically around 100–500 nm. On the other hand, EXOs originate from endosomes, with their sizes ranging between 40 and 100 nm (Figure 1).

However, the relevant nomenclature can often be confusing, leading the International Society of Extracellular Vesicles (ISEV) to propose the generic term EVs for vesicles released by cells [8]. Currently, EV subpopulations are classified based on their size as small (sEV) or medium/large (m/l EV) if they are less than or above 200 nm, respectively [8,9].

The knowledge about EVs is rapidly expanding and holds significant potential for various applications that continue to evolve. Their internal composition varying on the basis of physiological state makes them attractive as biomarkers for diagnosing, monitoring the progression of, and assessing the treatment responses for several diseases, particularly in cancer, neurology, metabolomics, and vaccine development [10,11,12]. However, the clinical potential of EVs is hindered by the lack of standardization in the methods for their isolation. The importance of obtaining high-quality extracts is crucial to preserving the integrity of the particles and their contents. Additionally, multiple cleanup methods are often required, increasing costs, time, and the need for technical expertise [13,14,15]. The observed profiles of EVs are deeply influenced by the adopted sample extraction procedures, further complicating the task.

Indeed, the development of reliable analytical methods for the rapid and cost-effective characterization of EVs and their internal cargo would significantly contribute to their clinical use [16]. In this context, considerable interest has been garnered by Matrix-Assisted Laser Desorption Ionization–time-of-flight/mass spectrometry (MALDI-TOF/MS), a technique that has demonstrated its ability to provide valid and accurate information from complex biological samples for diagnostic purposes [17,18], especially when a high throughput, accurate identification, and simple sample pre-treatment are required. MALDI is well suited for clinical practice, as it is user-friendly even for unskilled personnel, allowing for the rapid generation of spectra and profiles from the matrix under study. Particularly, the potential, limitations, and potential implementation strategies of MALDI analysis of exosomes have recently been thoroughly discussed [19], while applications of MALDI to the analysis of the EVs secreted by cancer cells have been reviewed [20]. Given the considerable and growing number of existing works focusing on MALDI characterization of the EVs released by human cells in various pathological “non-tumor” conditions, as well as by animal cells and microorganisms, this work aims to comprehensively review the related literature.

## 2. EV Isolation Techniques

Extracellular vesicle characterization suffers from a lack of standardized strategies for their profiling, limiting developments in clinical diagnostics [9]. To facilitate their study and increase knowledge, it is essential that EVs are specifically isolated and purified from the broad spectrum of cells, cellular debris, and interfering components that can be present in a sample, with the hope of minimizing losses, prior to mass spectrometry analyses.

With rapid advances in science and technology, numerous techniques have been developed for the isolation, purification, and preconcentration of the EVs released into biological fluids, allowing for the study of their lipidomic profile and protein expression through mass spectrometry analyses. These studies aim to define their molecular composition, biogenesis, and transported cargo. Recently, the main techniques for EV isolation have been reviewed [13,14,15].

Ultracentrifugation (UC), ultrafiltration (UF), and size exclusion chromatography (SEC) are conventional techniques based on particle size, widely used for isolating EV subpopulations (Figure 2(1)). Additionally, novel techniques such as polyethylene-glycol-based (PEG) precipitation, immunoaffinity (IA) capture, and microfluidics (MF)-based isolation methods (Figure 2(2)) have been developed [14,15], leveraging surface proteins for EV isolation. Each technique comes with its unique set of advantages and disadvantages.

### 2.1. Size-Based Methods

When a heterogeneous mixture undergoes centrifugation, suspended particles such as viruses, bacteria, subcellular organelles, and EVs can be separated based on their sedimentation coefficients. These coefficients are related to the physical properties, density, and viscosity of the medium. Ultracentrifugation (UC), which typically involves forces of ≥100,000× *g*, allows for the fractionation of smaller bioparticles, including medium and small extracellular vesicles (Figure 2(1)).

UC is favoured due to its ease of use, requiring minimal or no sample pre-treatment. It also enables processing of varying sample volumes in a short time frame and at relatively low costs, aside from the initial equipment investment, which is generally amortized over time. For these reasons, UC is considered the gold standard technique for isolating EVs from diverse sources such as urine, serum/plasma, cerebrospinal fluid, milk, cell culture medium, and environmental samples [14,15].

Differential ultracentrifugation (dUC) (Figure 2(1Aa)) is a prevalent method for isolating and purifying EV subpopulations. This technique involves a series of cycles at different relative centrifugal forces and times. The efficiency of EV isolation via dUC is influenced by four main factors: acceleration, rotor type, solution viscosity, and the time required to achieve the desired pellet formation [21]. Each variable’s impact has been extensively discussed in the literature [14,15]. However, dUC can be laborious and time-consuming for recovering smaller EVs, such as exosomes, as numerous centrifugation cycles are often necessary for particle separation. The EVs are typically recovered from the supernatant or pellet between cycles, depending on the applied centrifugal force, and are then resuspended in a suitable solution for storage at −80 °C until further analysis.

Urine is a preferred biological fluid for obtaining appreciable quantities of high-purity EVs non-invasively. This is due to the ease of obtaining large volumes and the relatively low presence of debris and proteins in urine. For plasma/serum samples, a cleaning step is necessary before UC to remove major bioparticles and the addition of protease inhibitors to prevent exosomal protein degradation. However, dUC often faces challenges such as contamination and EV loss [14,15].

UC on a density gradient (Figure 2(1Ab,c)) offers an alternative technique for separating particles with similar densities but different masses in a less labor-intensive manner. A density gradient is created by layering solutions of decreasing density, typically based on carbohydrates, colloidal silica gels, or iodine compounds, in a centrifuge tube. This gradient can either be formed before centrifugation (zonal band UC, Figure 2(1Ab)) or during centrifugation (equilibrium density gradient or isopycnic, Figure 2(1Ac)). The EVs are then sedimented in discrete zones within the gradient based on their density, mass, and size relative to the suspension medium. These EVs can be retrieved by puncturing the tube at its lower end and collecting individual fractions using a fraction collector. Isopycnic UC involves centrifuging the sample and solution mixture until equilibrium is achieved, allowing the EVs to sediment based on their density [14,15].

Samples can be stripped of larger particles by forcing the fluids to pass through porous membranes, applying pressure through physical and/or mechanical means (Figure 2(1B)). The porous membranes used may have different molecular weight or size exclusion limits and are generally derived from modified natural cellulose or synthetic materials. Through this technique, it is possible to obtain high preconcentration values of EVs [22]. It is not uncommon for exosomal UF preparations to be significantly contaminated by floating non-exosomal humoral peptides such as alpha-1-antitrypsin and albumin [23]. For cell-free samples such as urine, serum, cerebrospinal fluid, and cell culture medium, nanomembrane concentrators are a valid and effective alternative to UC for quickly and successfully isolating urinary exosomes from small volumes, such as 0.5 mL of urine [23].

Another technique is size exclusion chromatography (SEC) or gel filtration (Figure 2(1C)), a size-based method that utilizes a resin-packed column. SEC offers high-purity and functionally intact EV isolation due to gravitational flow and minimal sample handling. It also provides excellent reproducibility, although its long run times can limit its use for high-throughput applications.

Rood et al. [24] compared the three size-based methods, i.e., UF, UC, and SEC, alone and combined with each other to obtain amounts of urinary microvesicles useful for the detection of protein biomarkers of nephrotic syndrome using MALDI-TOF-TOF mass spectrometry, finding that UF and UC alone are less efficient for microvesicle enrichment than UC combined with SEC (UC-SEC). In fact, in the latter case, the preparations were highly microvesicle-enriched.

Comparison studies have shown that dUC coupled with UF can yield EV extracts suitable for characterizing the total protein content and particle size distribution. For instance, Piotrowska et al. successfully identified 62 protein markers in EVs from Gram-negative rods of the Enterobacteriaceae family using MALDI-TOF/TOF-MS proteomic analysis [25]. It allows for a high preconcentration of EVs, although some UF preparations may be contaminated with non-exosomal peptides. To address this, tangential flow filtration can be used to remove contaminants after an initial filtration step, excluding particles larger than 1 µm. The filtrate, containing the exosomes, can then be further concentrated and fractionated using filters with specific pore sizes [26].

UF, compared to UC, offers lower costs and faster processing and does not require specialized equipment [27]. However, loss of EVs may occur due to membrane attachment, vesicle entrapment, or filter media clogging [28]. Nanomembrane concentrators with brief centrifugation periods are also effective alternatives to UF for isolating urinary exosomes from cell-free samples such as urine, serum, cerebrospinal fluid, and cell culture medium [27].

In conclusion, a combination of different techniques such as differential centrifugation followed by ultrafiltration or size exclusion chromatography enables efficient isolation and purification of EVs, allowing for their subsequent characterization in terms of their composition and functional properties. These methodologies have significantly advanced our ability to study the role of EVs in various physiological and pathological processes.

### 2.2. Novel Techniques

The most cutting-edge techniques for isolating, purifying, and preconcentrating extracellular vesicles (EVs) include polyethylene-glycol-based precipitation (PEG), immunoaffinity capture (IA) methods, and microfluidic technology (MF) (Figure 2(2)).

PEG-based precipitation (Figure 2(2A)) involves the use of a solution to induce the formation of a vesicle aggregate trapped in the polymer in large quantities. On the other hand, IA-based techniques (Figure 2(2B)) offer a robust platform for selectively isolating EVs. These techniques utilize antibodies that target exosomal surface proteins, which are bound to solid media, enabling the selective capture of extracellular vesicles. Additionally, specific antibody-mediated binding chips (Figure 2(2C)) can be employed to efficiently capture EVs based on a combination of their immunoaffinity, size, and density properties. To achieve this, the devices can be equipped with filter pore sizes (<500 nm) to isolate the EVs without causing aggregation.

For instance, the microfluidic chip developed by Han et al. (2021) [29] symmetrically combines two layers of polymethyl methacrylate (PMMA) with serpentine channels and a nanoporous hydrophobic membrane (~100 nm diameter) made of rail-etched polycarbonate (PCTE) in the middle. The uniform pore-sized PCTE membrane enables efficient cleaning of protein contaminants (>97%) and allows for the collection of exosomes trapped in the microfluidic chip through reverse elution with high recovery rates (>80%) directly from human blood plasma. Compared to the “gold standard” of ultracentrifugation (UC) for exosome isolation and purification, the size-dependent microfluidic-chip-based protocol demonstrates higher efficiency in protein cleaning and adequate recovery rates for exosomes. The entire process is completed in under 3 h, which is notably shorter than UC. Given that microfluidics-based methods can be automated, this method holds promise for clinical applications in large cohorts.

Recently, Hua et al. [30] developed a microfluidic device based on dual-flow filtration for highly efficient separation and enrichment of EXOs.

Ye et al. [31] introduced an innovative approach for isolating and detecting small extracellular vesicles (sEVs) from blood using the EXODUS system (Figure 3). This system is a dual-membrane nanofiltration system that integrates periodic negative pressure oscillation (NPO) and double-coupled ultrasonic harmonic oscillation (HO), combined with MALDI-TOF/MS and LC-ESI-MS/MS, for sensitive detection. The EXODUS system achieves label-free isolation with a relatively high recovery and purity of EVs from a small volume of plasma (20 μL). Comparative studies with commonly used methods such as PEG-based precipitation and ultracentrifugation highlight the efficiency and advantages of the EXODUS system [32]. The combination of the EXODUS system for isolation and MALDI-TOF MS for detection is proposed as a clinical translation method, offering speed and high-throughput capabilities that could significantly impact biomarker detection and screening in clinical settings [31,32].

Exosome precipitation is characterized by its ease of implementation, lack of specialized equipment requirements, and its ability to handle large and scalable sample capacities [31]. However, its limitations include the co-precipitation of non-exosomal contaminants such as proteins and polymeric materials, as well as the need for extensive pre- and post-cleanup steps, leading to longer execution times. Additionally, the varying viscosity and matrix of the sample present challenges, making standardization of the precipitation protocols difficult [14,33].

On the other hand, immunoaffinity-based techniques excel in isolating and purifying EVs of diverse origins. However, they are associated with high reagent costs, limited capacity, and low yields. Heterogeneous samples can also hinder immune recognition, as the antigenic epitope may be blocked or masked [34]. Microfluidics-based techniques offer advantages such as speed, affordability, portability, and ease of integration. Yet they lack standardization, large-scale testing on clinical samples, and method validation and have moderate to low sampling capabilities [35].

Each isolation technique targets specific properties or characteristics of the extracellular vesicles, such as density, size, or immunoaffinity. However, a consensus on the optimal technique has yet to be reached [36]. Parameters such as the degree of purity of the isolated EVs, costs, working hours, and technical training required for the separation phases play significant roles in the selection of the appropriate method.

## 3. MALDI/MS-Based Biomarker Analysis of EVs from Non-Cancer Cells

The recent development of various “omics” fields has induced significant advances in the search for non-invasive biomarkers based on the analysis of body fluids for a wide spectrum of diseases. Extracellular vesicles can encapsulate a wealth of information. However, the methods used for their isolation can greatly impact the downstream analysis. Therefore, the techniques for EV preparation play a critical role in ensuring the accuracy of protein and lipid analysis, especially when employing mass spectrometry for diagnostic applications.

For numerous biomedical investigations, the MALDI ionization source stands out for its ability to analyze biomolecules within a mass range of up to 300–500 kDa with high sensitivity (ranging from pico- to femtomole levels). During MALDI, a UV laser pulse desorbs a matrix/analyte–particle cloud from the sample co-crystalline matrix/solid solution deposited on a metal target. This process involves the transfer of protons from the matrix ions, primarily responsible for generating the analyte ions that are then transferred to the mass analyzer for detection. The high transmission efficiency of mass analyzers allows for precise and sensitive determination of the mass-to-charge ratio (*m*/*z*) of the ions generated in the source. Additionally, the ionization process can tolerate millimolar salt concentrations and is considered a soft ionization technique, resulting in minimal fragmentation. These characteristics make MALDI suitable for analyzing complex mixtures.

However, MALDI also presents some challenges, including a matrix background that is highly dependent on the matrix material used, which can be problematic for samples below a mass of 1000 Da. Additionally, there is the potential for photochemical degradation of the analyte molecules by UV/IR laser radiation [37].

Mass analyzers interfaced with MALDI sources come in various types. Time-of-flight (TOF) mass analyzers, including advancements in the linear mode, reflectron, and orthogonal modes, have improved the mass resolution and accuracy [38]. Quadrupole ion traps, Fourier transform ion cyclotron resonance (FTICR) instruments, and more recently Orbitrap mass analyzers have also met the requirements for high-resolution, accurate, and sensitive mass spectrometry in analyzing the complexity of biological samples such as the proteome and metabolome [39,40,41,42]. Modern MS instruments are often hybrids, combining two or more different mass analyzers (represented by TOF/TOF, quadrupole–TOF, and quadrupole–orbitraps), allowing for rapid, accurate, and precise mass determination of the precursor and product ions—an essential goal in MS [37].

For proteome determination, isolated, highly purified, and preconcentrated EVs are typically digested with trypsin, and approximately 1 μL is deposited on a target plate along with the matrix (Figure 4). Although this approach is rapid, a more thorough method based on preliminary separation using SDS gel electrophoresis followed by gel digestion of the proteins extracted from the vesicles is often preferred. This more extensive process ensures the cleanliness of the spotted samples, which is crucial for the accuracy of MALDI/MS protein characterization. During analysis, the mass spectrometer (normally) operates in positive ion mode over a normal mass range of *m*/*z* 700–3000, with the mass resolution setting typically at R = 100,000. Each measurement generates an average MALDI/MS spectrum with numerous *m*/*z* values. The differential composition of the *m*/*z* peaks is then compared with protein spectra stored in a database and directly associated, with each associated protein assigned a variable level of reliability. The accuracy of most MALDI/MS instrumentation ranges between 98% and 99%, establishing them as a gold standard in clinical diagnostics.

In the case of vesicular lipidome analysis, the total lipids are extracted using organic solvents and directly deposited on the metal target, with or without mixing with the matrix (Figure 4). The instrument can also operate in negative ion mode over a mass range of 400–2000 *m*/*z*, offering high mass resolution. For instance, Peterka et al. [43] characterized the lipidomic component of extracellular vesicles (EVs) isolated from the human plasma of healthy donors using various mass spectrometry techniques. MALDI provided insights into classes of anionic lipids such as sulfatides. Comparing the lipid classes in the EVs and plasma, significant differences were found in triglycerols, diacylglycerols, phosphatidylcholines, and lysophosphatidylcholines, while sphingomyelins, phosphatidylinositols, and sulfatides exhibited relatively similar profiles in both the EVs and plasma. In another study, Banliat et al. [44] demonstrated that EVs from oviduct fluid influence the phospholipid composition of bovine embryos developed in vitro.

To illustrate the significant implications of lipid component studies of extracellular vesicles for clinical applications, consider the work of Madonna et al. [45] on the mechanisms underlying the therapeutic effects of stem/progenitor cells, including adipose-tissue-derived mesenchymal stromal cells (AT-MSCs).

### 3.1. Applications

#### 3.1.1. Various Applications

An increase in the number of EVs in cellular fluids is observed in all pathological conditions. Therefore, all studies aimed at improving the preparation of these small particles and their characterization for the identification of specific biomarkers are of great interest, and MALDI/MS can represent a valuable tool. The sensitivity of the instrumental technique in detecting molecular signals can significantly reduce the time and workload required to confirm a patient’s health status given the multiple implications of EVs. MALDI characterization of the extracellular vesicles released by human cells in various “non-cancerous” pathological conditions, and, to a lesser extent, by animal cells and microorganisms, is the aim pursued by many researchers. Table 1 shows a summary of the objectives and MALDI/MS analysis of EVs released under "non-cancerous" conditions from human cells, animals and microorganisms.

In this regard, Nguyen et al. [46] used a MALDI source interfaced with a Fourier transform ion cyclotron resonance mass spectrometer (FTICR-MS) to detect signals of the presence of EVs in biological fluid and to demonstrate how their detection can be useful in clinical practice. Human-serum-enriched exosomes in a mass range of *m*/*z* 1000–20,000 were obtained using simple centrifugation, which were subsequently analyzed using MALDI-FTICR-MS, producing a distinctive protein around *m*/*z* 7766, identified by the database as platelet factor 4 (PLF4). The MALDI-TOF spectra of the EXOs from patients with different liver disease states showed a very abundant common peak, attributable to PLF4, along with several additional minor peaks. PLF4 was proposed by the authors as a molecular signal of extracellular vesicles and MALDI/MS as a valid tool for their determination in plasma fluid correlated with the disease.

Sample cleanliness plays a key role in protein and lipidomics analysis. Therefore, great advantages are obtained by subjecting samples to instrumental analysis that are more representative of each individual extravesicular subpopulation. Burkova et al. [47,48] prepared crude nanovesicles from a variety of sources, including the placenta (normal pregnancy), using a gold standard procedure followed by filtration and affinity chromatography on Sepharose containing immobilized antibodies against the CD81 exosome surface protein. Identification of the exosomal proteins was performed through MALDI mass analysis of the tryptic protein hydrolysates after SDS-PAGE and 2D electrophoresis. The EXOs (30–100 nm) from the human placenta that have an affinity for CD81 were shown to contain more than 27 different peptides and small proteins of 2–10 kDa, while the well-extrapurified exosomes contained only 11–13 different proteins: CD9, CD81, CD-63, hemoglobin subunits, interleukin-1 receptor, annexin A1, annexin A2, annexin A5, cytoplasmic actin, alkaline phosphatase, serotransferrin, and probably human serum albumin and immunoglobulins.

MALDI sample preparation techniques must be able to selectively isolate the sEVs present in different fluids based on their size, density, and origin, and recoveries are important for this type of investigation. Lower recoveries can adversely affect the downstream proteomics, lipidomics, and genomics analyses, not to mention the sample purity. In fact, the profiles of the mass spectra are generated through the acquisition of ions of different *m*/*z* generated by the MALDI ionization of the samples deposited on the target plate. Different samples correspond to different spectra. The reproducibility of the instrumental response for the purposes of clinical research/the treatment of specific diseases is related to the sample. Pure urinary EXOs were isolated using differential centrifugation by Saraswat et al. [49] to define their N-glycoproteomic profile, useful for detecting biomarkers of urological and other diseases. For sample enrichment of the tryptic hydrolysates of the protein samples prior to mass spectrometry analysis, they used an IA-capture-based technique or gel filtration chromatography. In total, in this study, 66 unique non-modified N-glycan compositions and 13 sulfated/phosphorylated glycans were found using MALDI/MS [49].

Plasma exosomes were extracted using ultracentrifugation from a control group of healthy subjects and a group of patients with viral myocarditis (VMC) in the search for biomarkers of the disease by Zhao et al. [50]. Difference gel electrophoresis (DIGE) technology was used to isolate the total proteins, which were subsequently identified using MALDI-TOF/TOF mass spectrometry and verified using ELISA. A total of ten VMC-related proteins were detected, and RBP4 (Retinol Binding Protein-4) was suggested as a potential specific biomarker for early screening and diagnosis of the pathology.

To study the protein profile of the serum exosomes of children with coronary artery aneurysm (CAA) caused by Kawasaki disease (KD), Xie et al. [51] used two-dimensional electrophoresis (2-DE). Thirty-two differentially expressed proteins were identified (eighteen up-regulated and fourteen down-regulated) from the patients’ serum exosomes using MALDI-TOF-TOF analysis. From the comparison with the healthy controls, the authors validated the expression levels of four proteins (TN, RBP4, LRG1, and APOA4) via Western blotting analysis, establishing a complete proteomic profile of the serum exosomes of the children with CAA caused by KD and providing further information on the mechanisms of CAA caused by KD.

Song et al. [52] believe that a detailed characterization of the urinary exosomal components in healthy individuals is essential for investigating the urinary tract. Since alterations of sialylation patterns have been implicated in various disease states, ion exchange chromatography, microfluidic capillary electrophoresis (CE), and MALDI/MS have been adopted to resolve the positional isomers of sialic acids, identifying 219 N-glycan structures in human urinary EXOs.

Korenevsky et al. [66] studied the MALDI-TOF mass spectrometric protein profile of microvesicles produced in vitro by the natural killer cell line NK-92. A total of 986 proteins with a variety of functions, from extracellular and intracellular regulatory signals to lipids and vitamins, have been identified in the microvesicle lysate. The data obtained expand the existing knowledge on remote cell communication and indicate new mechanisms of interaction between natural killer and target cells.

Sedykh et al. [54,55] purified horse milk EXOs using affinity chromatography. Exosomal proteins were identified before and after gel filtration using MALDI MS and MS/MS spectrometry of SDS-PAGE-derived tryptic protein hydrolysates and 2D electrophoresis.

#### 3.1.2. Hormonal and Metabolic Disorders

Diabetes is a chronic condition that affects approximately 425 million people worldwide [67]. It is characterized by high blood glucose levels, which can lead to the development of severe, and even lethal, complications. The disease is related to insulin, a hormone produced by the pancreas, which regulates blood glucose levels. The causes of diabetes may be related to the reduced availability of insulin and/or abnormal responses of the body’s cells to it. Communication between organs is critical for maintaining glucose homeostasis. Recently, several studies have highlighted the role of exosomes in the development of diabetes and associated complications affecting the blood vessels, kidneys, liver, eyes, skeletal muscle, and nervous system [56]. The damage caused by diabetes to various organs can be cumulative and severe; for example, the kidneys can present functional alterations, leading to chronic kidney disease, which may eventually require dialysis or organ transplant.

Nano-liquid chromatography coupled offline with MALDI-TOF/MS/MS was applied by Kaminska et al. [56] for the proteomic analysis of the urinary EVs from diabetic patients without renal failure (NRF) and with renal failure (RF). EVs were found to be abundant in the urine of all the patients enrolled in the trial. The density and size of the urinary EVs reflect the deterioration of renal function and therefore can be considered potential biomarkers of renal damage. Albumin, uromodulin, and several unique proteins related to RF were also detected in the EV fraction of the RF patients.

Gu et al. [57] conducted a study based on mass spectrometry and bioinformatics analysis to identify potential biomarkers. Using the MASCOT software search engine, they identified 22 differential proteins, and MASP2, CALB1, S100A8, and S100A9 were selected as potential biomarkers of early diabetes mellitus.

The study of cellular microvesicles using MALDI-TOF/MS/MS has also proven useful for studying galactosemia, a rare autosomal recessive metabolic disorder characterized by the inability to metabolize galactose (GALT) [68,69]. The mainstay of treatment for galactosemia is a GALT-free diet as soon as the disease is suspected. However, serious complications of the central nervous system and ovaries can still occur in the long term because GALT, even when eliminated from the diet, can be synthesized endogenously by cells. EVs were therefore considered biomarkers of the disease. Staubach et al. [68] used urinary exovesicles and Tamm–Horsfall glycoprotein, expressed exclusively in the kidneys, as non-invasive cellular material to study GALT deficiency. By applying matrix-assisted laser ionization mass spectrometry to the permethylated N-glycans released by PNGaseF, they demonstrated that GALT deficiency is associated with substantial glycan modifications. In particular, they found that complex-type N-glycans are pre-existing in subjects suffering from galactosemia, compared to healthy subjects, where they are of the high-mannose (M) type. Furthermore, these modifications concerned exclusively the exosomal proteins and not the glycoprotein expressed by the kidney, which always showed a predominant glycosylation of the high-mannose type (M6).

#### 3.1.3. Neurodegenerative Diseases

Neurodegenerative diseases affect one in six people and are now the most common pathological condition in Western countries [70,71]. Unfortunately, this statistic is likely to increase due to the aging of the population. Unlike other major diseases, such as cardiovascular, respiratory, infectious diseases, and some cancers, there are no effective preventive measures and therapeutic treatments available [72,73,74]. Additionally, precise diagnosis can be challenging, particularly in the early stages, due to the similar clinical features among the different forms.

Alzheimer’s disease (AD) is the most prevalent form of progressively disabling degenerative dementia, with onset predominantly in the presenile age group (over 65 years) [75]. Although the rate of progression can vary, the average life expectancy after diagnosis ranges from three to nine years [68]. It is estimated that 1 in 85 people worldwide will be affected by AD by 2050 [76]. Despite extensive studies, the cause and progression of AD are still not well understood. Research indicates a close association of the disease with the amyloid plaques and neurofibrillary clusters found in the brain, yet the underlying cause of such degeneration remains unknown [77], and there is currently no available treatment. Given these challenges, accurate identification methods are crucial to mitigating the impact of the disease.

Wu et al. [59] developed a material based on the CaTiO3/Al3 + /Pr3 + /Sm3+ nanocomposite for the enrichment of blood exosomes. Through proteomic analysis, they demonstrated that serum exosomes from AD patients and healthy controls exhibit variability in the expression levels of certain proteins. Many of these proteins are associated with an inflammatory response, such as the precursor complement factor H-related protein 1 and a homologue of factor H. Furthermore, the P-component of serum amyloid (SAP) was found to be upregulated nearly 2-fold in AD patients compared to healthy controls. Consequently, it has been suggested that SAP may be related to the pathogenesis of AD. Extracellular vesicles originating from the central nervous system contain biological materials specific to the cells and the cellular state of CNS organs (such as the brain, spinal cord, cerebellum, and brainstem). Consequently, these EVs can cross the blood–brain barrier, enter the bloodstream, and be detected in other bodily fluids, including saliva, tears, and urine. Similarly, toxic substances and misfolded forms of amyloidogenic proteins can spread to recipient cells in the CNS. The method developed by Wu et al. [59] has demonstrated potential for application in clinical practice due to its practicality and speed. However, some technical issues still need standardization for the identification and quantification of biomarkers for neurodegenerative diseases through EV isolation and MALDI-TOF analysis, such as determining the best surface markers for isolating cell-type-specific microvesicles and validating their cellular origin.

#### 3.1.4. Immune System, Infectious Disease, and Vaccines

The immune system plays a crucial role in defending the body against chemical, traumatic, or infectious insults, thereby maintaining its integrity [78]. The ability of the immune system to induce an inflammatory process is essential for preserving biological functions and cell survival. However, this ability is not always beneficial. Autoimmune diseases or allergic reactions can result in abnormal recognition of external agents and excessive, altered inflammatory responses. These diseases are often chronic and challenging to treat, necessitating ongoing research to understand their mechanisms of development and progression, as well as to identify more effective therapeutic treatments to combat or limit their damage.

Refractory nephrotic syndrome (RNS) is an immune-related kidney disease with poor clinical outcomes. Nephrotic syndrome is typically caused by damage to the groups of small blood vessels in the kidneys responsible for filtering waste and excess water from the blood. Neprilysin, aquaporin-2, and podocalixin were identified as urinary microvesicular biomarkers of nephrotic syndrome detected using MALDI-TOF/TOF mass spectrometry by Rood et al., who proposed their use in predicting the clinical course of the disease [24].

Kawasaki syndrome (KD), also known as mucocutaneous lymph node syndrome, is an infantile vasculitis of the medium and small arteries of an autoimmune nature that primarily affects the coronary arteries. Xie et al. [51] studied the protein profile of serum exosomes from children with coronary artery aneurysms (CAAs) caused by KD. Thirty-two differentially expressed proteins were identified (eighteen up-regulated and fourteen down-regulated) in the serum exosomes of patients with CAA and compared with those of healthy controls. The expression levels of TN, RBP4, LRG1, and APOA4 were validated using Western blotting. Protein–protein classification and network analyses revealed that these four proteins are associated with multiple functional groups, including host immune response, inflammation, apoptotic processes, developmental processes, and biological adhesion processes. These findings provide valuable insights into the mechanisms of CAA caused by KD.

Dry eye syndrome (DES) is a multifactorial ocular surface disease characterized by a loss of tear film homeostasis, accompanied by ocular symptoms. Its etiological factors include tear film instability, hyperosmolarity, ocular surface inflammation and damage, and sensorineural abnormalities [79]. The prevalence of DES varies widely worldwide (from 6.5% to 52.4%), with higher rates observed in postmenopausal women, people of Asian ethnicity, and with increasing age (between 2.0% and 10.5% per decade).

Tears are a biological fluid with diagnostic potential for eye diseases. The extracellular vesicles detected in tears are considered promising sources for non-invasive liquid biopsy. Zhang et al. [60] developed a mass-spectrometry-based strategy to analyze the peptidome/proteome profiles of tears and extracellular vesicles for the rapid diagnosis of ocular syndromes. Nano-sized extracellular vesicles were isolated from the tears of both healthy control individuals and patients with DES. Lacrimal extracellular vesicles and tears were characterized using MALDI-TOF analysis. Comparison of the discriminant peaks and extracellular vesicles between the tears proved to be an efficient method for screening potential DES biomarkers. The authors hope that their proposed approach to tear and EV fingerprinting will serve as a valuable tool for the rapid diagnosis of eye diseases and further research into their pathogenesis.

Extracellular vesicles can be released from all cells, including microbial cells. Characterizing these vesicles through omics sciences can provide medical signals, such as objective indications (biomarkers) of a patient’s health status and the responsible pathogens, allowing for their monitoring in a simple, accurate, and reproducible manner throughout the course of the disease.

The microbial EVs released into serum were studied by Zhao et al. [50] for biomarkers in viral myocarditis (VMC), an inflammatory disease of the heart muscle that is the leading cause of sudden death in men under 40 years of age. The plasma exosomes released by microorganisms were analyzed using MALDI-TOF/TOF mass spectrometry. Ten proteins—KRT2, KRT5, KRT9, KRT77, KRT78, AZGP1, HP, RBP4, CD5L, and C1QB—were found to be correlated with VMC overall, with RBP4 identified as a potential specific biomarker for early screening and diagnosis of the disease.

Paingankar et al. [60] suggested that exosomes could have important implications in the life cycle of the hepatitis virus (HEV).

Microbial infections involve complex interactions between microbes and the host, determining their consequences, such as commensalism, disease, or colonization, depending on the degree of damage caused over time. All pathogenic forms impair host immunity, with the severity of the disease largely dependent on the integrity of the host’s immune system. For instance, virulent pathogens may cause minimal or no symptoms in individuals with robust immunity, whereas avirulent pathogens can lead to severe symptoms in immunocompromised patients [80]. Pathogens have the capacity to acquire pathoadaptive traits that enhance their survival and ability to evade the immune system’s control.

The study of EVs addresses aspects relevant to both basic research and the fundamental mechanisms of life. Enhanced comprehension of cellular communication and regulation mechanisms enriches our knowledge for the practical development of effective strategies to control hypermutable pathogens that infect humans, animals, and plants, as well as contaminants in cell cultures and vaccines.

Acholeplasma laidlawii (class Mollicutes) is a mycoplasma, a small prokaryote capable of autonomous reproduction that is ubiquitous in its infection of higher eukaryotes. It serves as a major contaminant of cell cultures and vaccines. Medvedeva et al. [61] revealed that the development of antimicrobial resistance in this bacterium is associated with the secretion of extracellular vesicles. In an in vitro study, Mouzykantov et al. [62] demonstrated that mycoplasma extracellular vesicles can infiltrate eukaryotic cells and modulate immunoreactivity.

The role of the exosomes released by macrophages infected with Mycobacterium avium subsp. paratuberculosis was examined in intercellular communication processes by Wang et al. [63]. They observed changes in the proteomic profile and immune properties of the exosomes released by stimulated macrophages. The identified differentially expressed proteins were closely associated with the cytoskeleton, protein synthesis and processing, and the inflammatory response [64].

Viruses of the Paramyxoviridae family are responsible for diseases such as measles and mumps, both preventable through vaccination. Besides the proteins encoded by the viral genome, vaccines may also contain host cell proteins due to EVs often being co-purified with viruses due to their size [65].

## 4. Conclusions

In conclusion, the characterization of the EVs released in non-cancer diseases through MALDI/MS represents a promising avenue of research, with significant implications for clinical practice. The study of EVs has evolved from considering them to be cellular waste disposers to recognizing their crucial roles in intercellular communication within healthy and diseased tissues. In various non-cancer pathologies, including autoimmune disorders, neurodegenerative diseases, and renal conditions, EVs have emerged as key players in disease progression and pathogenesis.

The application of MALDI/MS in this field has offered valuable insights into the proteomic profiles of EVs, shedding light on potential biomarkers for disease diagnosis, prognosis, and treatment response. Studies focusing on the identification of specific proteins, lipids, and nucleic acids within EVs have provided a deeper understanding of the underlying mechanisms of these diseases.

However, challenges remain, particularly in standardizing the isolation protocols, integrating analytical methods into clinical workflows, and addressing the heterogeneity of EV populations. Despite these hurdles, the increasing number of publications utilizing MALDI/MS for EV characterization underscores its growing importance in non-cancer disease research.

Moving forward, further advancements in MALDI/MS technology, coupled with refined isolation techniques and comprehensive bioinformatics analyses, hold the potential to revolutionize the field of non-cancer disease diagnostics and personalized medicine. Continued interdisciplinary collaborations between clinicians, researchers, and technologists will be pivotal in harnessing the full potential of EV characterization for improving patient outcomes and advancing our understanding of disease pathophysiology.

## Figures and Tables

**Figure 1 ijms-25-04490-f001:**
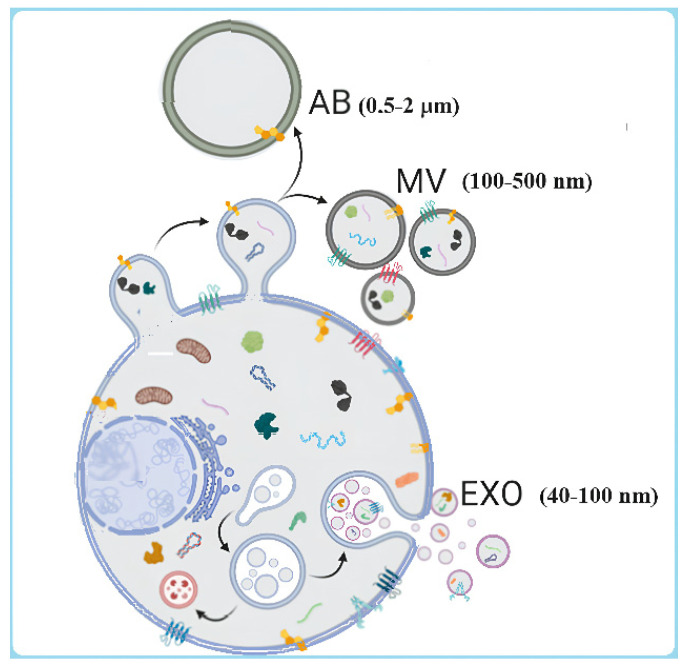
Biogenesis and subpopulations of EVs. Apoptotic bodies (ABs) are released by the blebbing of an apoptotic cell membrane (0.5–2 μm); microvesicles (MVs) are shed from the outward budding of the plasma membrane (100–500 nm); and exosomes (EXOs) are formed when multivesicular bodies fuse to the plasma membrane and release intraluminal vesicles (40–100 nm).

**Figure 2 ijms-25-04490-f002:**
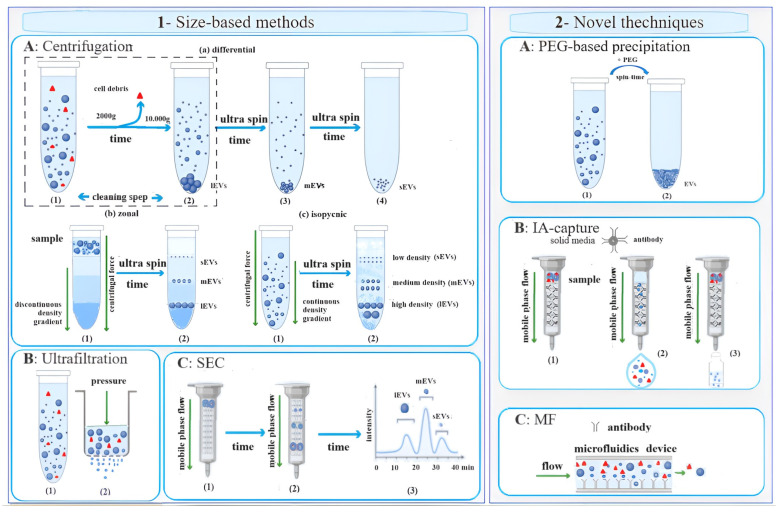
Conventional (1) and novel (2) methods of EV isolation. Conventional methods include centrifugation (1A), ultrafiltration (1B), and size exclusion chromatography (SEC) (1C). Centrifugation separates EV subpopulations according to their sedimentation coefficients, and they can be differential (1Aa), zonal (1Ab), or isopycnic (1Ac). Ultrafiltration uses a filter of a specific pore size that allows the filtrate to be enriched with vesicles of the desired size. SEC uses a porous stationary phase to differentially elute biofluid molecules based on their size. New techniques (2) are an effective alternative to conventional ones. Microvesicles can be trapped in the network of a polyethylene glycol (PEG) polymer (2A) to be selectively precipitated. Immunoaffinity (IA) capture (2B) uses antibodies that target exosomal surface proteins to isolate a specific population of vesicles. Microfluidic (MF) technology (2C) uses chips with a specific antibody-mediated binding to capture EXOs efficiently. 
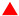
 = cell debris; 
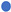
 = EV.

**Figure 3 ijms-25-04490-f003:**
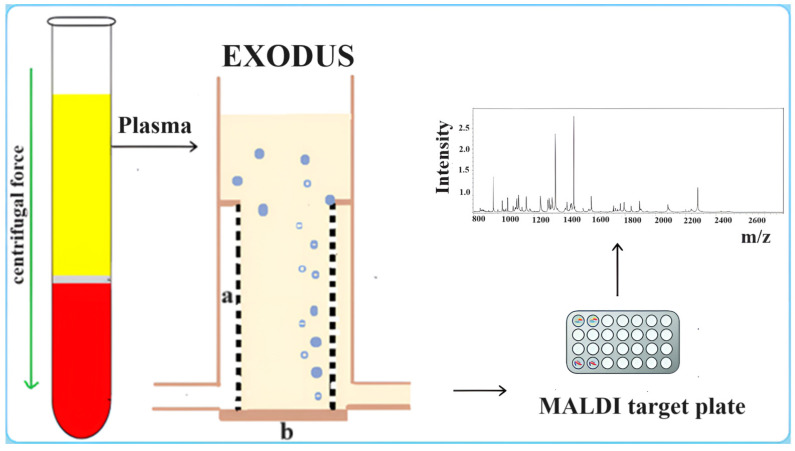
Centrifugation of blood (yellow: plasma; red: corpuscular part) and purification of EVs from plasma impurities by means of the EXODUS device, consisting of nanoporous membranes (a) and coupled harmonic resonators (b) managed by an integrated console. MALDI-TOF analysis for fingerprinting of EVs.

**Figure 4 ijms-25-04490-f004:**
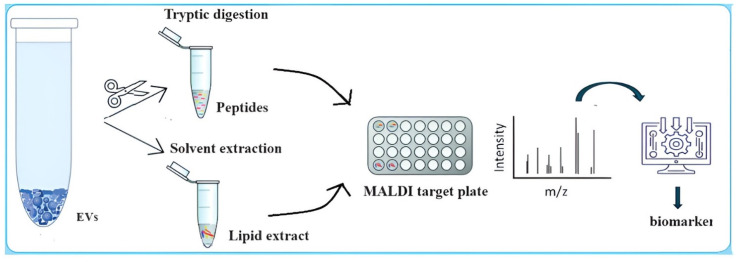
Scheme of the main steps of a MALDI/MS-based proteomic and lipidomics analysis of EVs for the software identification of a biological marker.

**Table 1 ijms-25-04490-t001:** Summary of the objectives and MALDI/MS analysis and characterization of EVs released in non-cancer diseases.

Authors	Objective	Techniques	Ref
Peterka et al.	Lipidomic characterization of exosomes isolated from human plasma.	Ultra-high-performance supercritical fluid chromatography–mass spectrometry (UHPSFC/MS), ultra-high-performance liquid chromatography–mass spectrometry (UHPLC/MS), and MALDI/MS.	[43]
Banliat et al.	Evaluation of the changes induced by oviduct fluid extracellular vesicles on embryo phospholipids.	Intact cell MALDI-TOF/MS.	[44]
Madonna et al.	Understanding the mechanisms of the therapeutic effects of stem/progenitor cells, including adipose-tissue-derived mesenchymal stromal cells (AT-MSCs).	Two-dimensional gel electrophoresis (2-DE) with MALDI-TOF/TOF.	[45]
Nguyen et al.	Proteomic characterization of exosomes in biological fluids.	MALDI combined with Fourier transform ion cyclotron resonance mass spectrometry (FTICR/MS).	[46]
Burkova, et al.	Search for protein biomarkers in human placenta exosomes.	MALDI/MS and MALDI/MS/MS of protein tryptic hydrolysates from SDS-PAGE and 2-DE.	[47,48]
Saraswat et al.	Investigation of the role of N-glycoproteome of urinary exosomes.	Collision-induced dissociation–MS/MS (CID–tandem MS) and MALDI/MS.	[49]
Zhao et al.	Biomarker screening for viral myocarditis through proteomic analysis.	MALDI-TOF/TOF mass spectrometry, validation using ELISA analysis.	[50]
Xie et al.	Comprehensive proteomic profile of serum exosomes from children with coronary artery aneurysms caused by Kawasaki disease.	2-DE with MALDI-TOF/TOF/MS.	[51]
Song et al.	N-glycan and sulfated N-glycan compositions in urine EVs for noninvasive investigations into the pathophysiological states of the urinary system.	Capillary electrophoresis–mass spectrometry (CE-MS), MALDI/MS, and capillary liquid chromatography–tandem mass spectrometry (LC-MS/MS).	[52]
Korenevsky et al.	Investigation on distant communications of cells and their regulatory mechanisms through the proteomic study of microvesicles derived from THP-1 monocyte cells.	MALDI/MS.	[53]
Sedykh et al.	Morphology and the protein content of major horse milk exosomes.	MALDI/MS and MS/MS spectrometry.	[54]
Sedykh et al.	Preparation of crude vesicles from horse milk for analysis of peptides and small proteins.	Standard methods of centrifugation, ultracentrifugation and gel filtration. Extra-purification with affinity chromatography on anti-CD81-Sepharose. Detection using MALDI-TOF/MS.	[55]
Kaminska et al.	Check the relationship between the density of urinary EVs, their size distribution, and the progress of early renal damage in type 2 diabetic patients (DMt2).	MALDI-TOF-MS/MS.	[56]
Gu et al.	Search for potential urine biomarkers for the diagnosis of prediabetes and early diabetic nephropathy.	Characterization using transmission electron microscopy (TEM), nanoparticle tracking analysis (NTA), and Western blotting of the tumor susceptibility gene product TSG101. Two-dimensional DIGE (2D-DIGE) with MS analysis.	[57]
Wu et al.	CaTiO_3_/Al^3+^/Pr^3+^/Sm^3+^ nanocomposite was synthesized and applied for highly selective and efficient separation of exosomes.	2-DE with MALDI TOF/TOF/MS.	[58]
Zhang et al.	To analyze peptidome/proteome profiles of tears and EVs for rapid dry eye diagnosis.	MALDI TOF/MS.	[59]
Paingankar et al.	To identify the host cellular factors that interact with Hepatitis E virus’ 5′ and 3′ untranslated regions (UTRs).	RNA pull-down and matrix-assisted laser desorption/ionization (MALDI) TOF.	[60]
Medvedeva et al.	To compare the cellular and vesicular proteomes of A. laidlawii strains with differing susceptibility to melittin (an antimicrobial peptide from bee venom).	2-DE with MALDI-TOF/TOF MS.	[61]
Mouzykantov et al.	To compare the genome profiles of ciprofloxacin-resistant A. laidlawii strains PG8r1 and PG8r3 selected under different in vitro conditions when a ciprofloxacin-sensitive A. laidlawii PG8B strain was cultured at increasing concentrations of ciprofloxacin in a broth medium alone and with vesicles derived from the ciprofloxacin-resistant A. laidlawii PG8R10c-2 strain, respectively.	2-DE with MALDI-TOF/TOF/MS.	[62]
Wang et al.	To explore the mechanism underlying the molecular immune response of macrophages stimulated with exosome [(+)exosome] from macrophages after Mycobacterium avium (M. avium) infection and analyze the differential protein component of the exosome.	2-DE with MALDI TOF/TOF MS.	[63]
Wang et al.	To study the role of exosomes shed from Mycobacterium avium sp. paratuberculosis-infected macrophages in intercellular communication processes.	MALDI-TOF/TOF.	[64]
Sviben et al.	To identify which virus-coded proteins are present in measles and mumps virus virions and to try to detect which host cell proteins, if any, are incorporated into the virions or adsorbed on their outer surface and which are more likely to be contamination from co-purified ECVs.	MALDI-TOF/TOF-MS.	[65]

## Data Availability

No new data was created in this review. Data sharing is not applicable to this article.

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
