# Peer review of "Analysis and Characterization of the Extracellular Vesicles Released in Non-Cancer Diseases Using Matrix-Assisted Laser Desorption Ionization/Mass Spectrometry"

_ijms, 2024, doi:10.3390/ijms25084490_

Round 1

Reviewer 1 Report

Comments and Suggestions for Authors

I liked this work. I have only minor suggestions . Please check the figures they do not seem high quaility, in particular figure 2.

Combine 3.1.4 and 3.1.5

Add a summary table of your work and exosomes applications.

Author Response

We thank the reviewer for his positive evaluation. Below are our point-by-point answers:

A1: I liked this work. I have only minor suggestions. Please check the figures they do not seem high quaility, in particular figure 2.

Q1: As suggested, the quality of the figures has been improved.

A2: Combine 3.1.4 and 3.1.5

Q2: Done.

A3: Add a summary table of your work and exosomes applications.

Q3: As suggested, the table has been added in the text.

Reviewer 2 Report

Comments and Suggestions for Authors

Comments and Suggestions for Authors:

The manuscript “Characterization of Extracellular Vesicles Released in 2 Non-Cancer Diseases by MALDI-TOF/MS” by Antonella Aresta and Carlo Zambonin is well written and presented in a clear manner. The references used are old, authors are suggested to add new and updated references can be added. I do have some comments on the figures. The resolution of the figures and the labeling is not appropriate. The figure descriptions are too brief and do not provide sufficient information for the reader.

1.      The word “Characterization” is inappropriate in the title. Change it to some specific word or modify the title.

2.      Page 1, Line 18 and 23: “MALDI-TOF-MS” and “MALDI-TOF/MS”. Are both the words have same meaning. If yes, then abbreviation should be properly written.

3.      Page 2, Line 52: apoptotic cell membrane (0.5-2mm). This is incorrect. Please write the correct size with proper units.

4.      Figure 1: Proper labeling of the pathway components needs to done. The author should submit higher resolution figure.

5.      Figure 2: The resolution is very low. The figure can be submitted in higher resolution. Also, the labeling is not proper with label position is not appropriate. I suggest the authors expand their description in each of the figures.

6.      Figure 3: Explain the figure in legend.

7.      Figure 4: intensity and m/z graph does not imply the identification of biomarker. Rather, the further downstream steps using the software analysis can be used to find the biomarker. Delete the arrow showing identification of biomarker and expand the figure to add missing steps.

8.      Page 13, Line 562, 568, 576, 578: In the word “MALDI-ToF/MS”, ‘o’ is small. Please make it uniform in the manuscript.

9.      References: Some of the references do not have doi. Reference style should be uniform.

Author Response

The manuscript “Characterization of Extracellular Vesicles Released in 2 Non-Cancer Diseases by MALDI-TOF/MS” by Antonella Aresta and Carlo Zambonin is well written and presented in a clear manner. The references used are old, authors are suggested to add new and updated references can be added. I do have some comments on the figures. The resolution of the figures and the labeling is not appropriate. The figure descriptions are too brief and do not provide sufficient information for the reader.

A: We thank the reviewer for his positive evaluation. New references have been added. The resolution of the figures has been improved, and the captions have been revised, but the detailed description and comments have been left in the text.

Below are our point-by-point answers:

Q1: The word “Characterization” is inappropriate in the title. Change it to some specific word or modify the title.

A1: We think the word characterization is appropriate, but we agree that it might be a bit restrictive, so the title has been changed as follows “Analysis and characterization of extracellular vesicles   released in non-cancer diseases by Matrix-Assisted Laser Desorption Ionization/Mass Spectrometry”.

Q2: Page 1, Line 18 and 23: “MALDI-TOF-MS” and “MALDI-TOF/MS”. Are both the words have same meaning. If yes, then abbreviation should be properly written.

A2: Done.

Q3: Page 2, Line 52: apoptotic cell membrane (0.5-2mm). This is incorrect. Please write the correct size with proper units.

A3: Done.

Q4: Figure 1: Proper labeling of the pathway components needs to done. The author should submit higher resolution figure.

A4: Done.

Q5: Figure 2: The resolution is very low. The figure can be submitted in higher resolution. Also, the labeling is not proper with label position is not appropriate. I suggest the authors expand their description in each of the figures.

A5: Done.

Q6: Figure 3: Explain the figure in legend.

A6: Done.

Q7: Figure 4: intensity and m/z graph does not imply the identification of biomarker. Rather, the further downstream steps using the software analysis can be used to find the biomarker. Delete the arrow showing identification of biomarker and expand the figure to add missing steps.

A7: Done.

Q8: Page 13, Line 562, 568, 576, 578: In the word “MALDI-ToF/MS”, ‘o’ is small. Please make it uniform in the manuscript.

A8: Done.

Q9: References: Some of the references do not have doi. Reference style should be uniform.

A9: Doi numbers have been deleated because not required.

Reviewer 3 Report

Comments and Suggestions for Authors

An article titled "Characterization of extracellular vesicles released in non-cancer diseases by MALDI-TOF/MS" submitted for review in the International Journal of Molecular Sciences provides an overview of methods for the isolation of extracellular vesicles and their analysis using the MALDI MS method. The paper is interesting and well-written; however, I have a few minor comments:
1. The authors describe the use of not only the MALDI ion source with the TOF analyzer, but also with other analyzers, so I suggest replacing the MALDI-TOF/MS acronym appearing in the title and replacing it with the full name - "Matrix-Assisted Laser Desorption/Ionization Mass Spectrometry" Also, in the abstract for the article, please drop the use of the term analyzer type. Similarly, please verify the correctness of the use of the abbreviation TOF for the methods described in Section 3.1. In at least one case, the authors use the acronym TOF and FTICR interchangeably, which is confusing to the reader.

2. Are there any other methods to characterize EVs? If so please list them in the article and at least a brief description. The authors could also include a table comparing these methods in the paper.

3. Please make a table comparing the EVs isolation techniques described in chapter 2.

Author Response

An article titled "Characterization of extracellular vesicles released in non-cancer diseases by MALDI-TOF/MS" submitted for review in the International Journal of Molecular Sciences provides an overview of methods for the isolation of extracellular vesicles and their analysis using the MALDI MS method. The paper is interesting and well-written; however, I have a few minor comments:

A: We thank the reviewer for his positive evaluation. Below are our point-by-point answers:

Q1: The authors describe the use of not only the MALDI ion source with the TOF analyzer, but also with other analyzers, so I suggest replacing the MALDI-TOF/MS acronym appearing in the title and replacing it with the full name - "Matrix-Assisted Laser Desorption/Ionization Mass Spectrometry" Also, in the abstract for the article, please drop the use of the term analyzer type. Similarly, please verify the correctness of the use of the abbreviation TOF for the methods described in Section 3.1. In at least one case, the authors use the acronym TOF and FTICR interchangeably, which is confusing to the reader.

A1: Done.

Q2: Are there any other methods to characterize EVs? If so please list them in the article and at least a brief description. The authors could also include a table comparing these methods in the paper.

A2: The ionization source for the characterization of extracellular vesicle components can also be non-MALDI. Since the purpose of the review is a critical examination of the MALDI/MS techniques, we believe that citation and description of the other analysis techniques is not necessary. However, summary table has been added in the text.

Q3: Please make a table comparing the EVs isolation techniques described in chapter 2.

A3: We think the table is not necessary since Martins et al. recently produced “A review on comparative studies addressing exosome isolation methods from body fuids”  (https://doi.org/10.1007/s00216-022-04174-5) which reports a clear comparition of all EVs isolation techniques. This review is now cited in the text.

Reviewer 4 Report

Comments and Suggestions for Authors

1- The novelty of the work is in using a technique widely applied to very specific structures (EVs).

2- They focus on MALDI, without understanding why they don't talk about ESI.

3- Missing, in the affiliation, the city and the country.

4- In the introduction, taking into account its structure, the mention of EVs in microorganisms or plants is missing.

5- I miss references (lines 135, 146, 156, 182, 221, for example).

6- The paragraph on line 173 should be joined to the previous one, after reference 24.

7- On line 288, indicate that "the mass spectrometer (normally) operates in positive mode...". (Madler et al., 2012, Journal of the american society of mass spectrometry).

8- Other applications beyond biomedical ones.

9- Scientific names in italics (lines 543 and 550 for example).

10- Standardize the format of references.